# Multisensory Encoding, Decoding, and Identification

**Aurel A. Lazar**
Department of Electrical Engineering
Columbia University
New York, NY 10027
`aurel@ee.columbia.edu`

**Yevgeniy B. Slutskiy**[*]
Department of Electrical Engineering
Columbia University
New York, NY 10027
`ys2146@columbia.edu`

## Abstract

We investigate a spiking neuron model of multisensory integration. Multiple stimuli from different sensory modalities are encoded by a single neural circuit comprised of a multisensory bank of receptive fields in cascade with a population of biophysical spike generators. We demonstrate that stimuli of different dimensions can be faithfully multiplexed and encoded in the spike domain and derive tractable algorithms for decoding each stimulus from the common pool of spikes. We also show that the identification of multisensory processing in a single neuron is dual to the recovery of stimuli encoded with a population of multisensory neurons, and prove that only a projection of the circuit onto input stimuli can be identified. We provide an example of multisensory integration using natural audio and video and discuss the performance of the proposed decoding and identification algorithms.

## 1 Introduction

Most organisms employ a mutlitude of sensory systems to create an internal representation of their environment. While the advantages of functionally specialized neural circuits are numerous, many benefits can also be obtained by integrating sensory modalities [1, 2, 3]. The perceptual advantages of combining multiple sensory streams that provide distinct measurements of the same physical event are compelling, as each sensory modality can inform the other in environmentally unfavorable circumstances [4]. For example, combining visual and auditory stimuli corresponding to a person talking at a cocktail party can substantially enhance the accuracy of the auditory percept [5].

Interestingly, recent studies demonstrated that multisensory integration takes place in brain areas that were traditionally considered to be unisensory [2, 6, 7]. This is in contrast to classical brain models in which multisensory integration is relegated to anatomically established sensory convergence regions, after extensive unisensory processing has already taken place [4]. Moreover, multisensory effects were shown to arise not solely due to feedback from higher cortical areas. Rather, they seem to be carried by feedforward pathways at the early stages of the processing hierarchy [2, 7, 8].

The computational principles of multisensory integration are still poorly understood. In part, this is because most of the experimental data comes from psychophysical and functional imaging experiments which do not provide the resolution necessary to study sensory integration at the cellular level [2, 7, 9, 10, 11]. Moreover, although multisensory neuron responses depend on several concurrently received stimuli, existing identification methods typically require separate experimental trials for each of the sensory modalities involved [4, 12, 13, 14]. Doing so creates major challenges, especially when unisensory responses are weak or together do not account for the multisensory response.

Here we present a biophysically-grounded spiking neural circuit and a tractable mathematical methodology that together allow one to study the problems of multisensory encoding, decoding, and identification within a unified theoretical framework. Our neural circuit is comprised of a bank

---

[*]The authors' names are listed in alphabetical order.

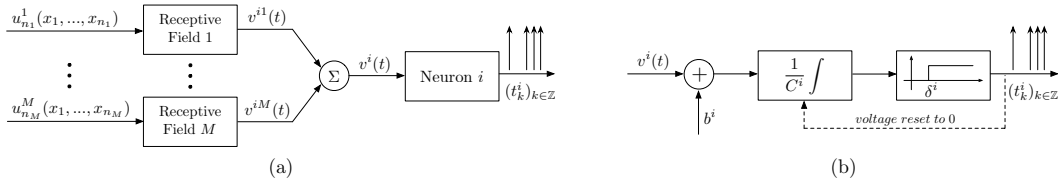

Figure 1: **Multisensory encoding on neuronal level.** **(a)** Each neuron $i=1,...,N$ receives multiple stimuli $u_{n_m}^m$, $m=1,...,M$, of different modalities and encodes them into a single spike train $(t_k^i)_{k\in\mathbb{Z}}$. **(b)** A spiking point neuron model, e.g., the IAF model, describes the mapping of the current $v^i(t)=\sum_m v^{im}(t)$ into spikes.

of multisensory receptive fields in cascade with a population of neurons that implement stimulus multiplexing in the spike domain. The circuit architecture is quite flexible in that it can incorporate complex connectivity [15] and a number different spike generation models [16], [17].

Our approach is grounded in the theory of sampling in Hilbert spaces. Using this theory, we show that signals of different modalities, having different dimensions and dynamics, can be faithfully encoded into a single multidimensional spike train by a common population of neurons. Some benefits of using a common population include (a) built-in redundancy, whereby, by rerouting, a circuit could take over the function of another faulty circuit (e.g., after a stroke) (b) capability to dynamically allocate resources for the encoding of a given signal of interest (e.g., during attention) (c) joint processing and storage of multisensory signals/stimuli (e.g., in associative memory tasks).

First we show that, under appropriate conditions, each of the stimuli processed by a multisensory circuit can be decoded loss-free from a common, unlabeled set of spikes. These conditions provide clear lower bounds on the size of the population of multisensory neurons and the total number of spikes generated by the entire circuit. We then discuss the open problem of identifying multisensory processing using concurrently presented sensory stimuli. We show that the identification of multisensory processing in a single neuron is elegantly related to the recovery of stimuli encoded with a population of multisensory neurons. Moreover, we prove that only a projection of the circuit onto the space of input stimuli can be identified. Finally, we present examples of both decoding and identification algorithms and demonstrate their performance using natural stimuli.

## 2 Modeling Sensory Stimuli, their Processing and Encoding

Our formal model of multisensory encoding, called the multisensory Time Encoding Machine (mTEM) is closely related to traditional TEMs [18]. TEMs are real-time asynchronous mechanisms for encoding continuous and discrete signals into a time sequence. They arise as models of early sensory systems in neuroscience [17, 19] as well as nonlinear sampling circuits and analog-to-discrete (A/D) converters in communication systems [17, 18]. However, in contrast to traditional TEMs that encode one or more stimuli of the same dimension $n$, a general mTEM receives $M$ input stimuli $u_{n_1}^1,...,u_{n_M}^M$ of different dimensions $n_m\in\mathbb{N}$, $m=1,...,M$, and possibly different dynamics (Fig. 1a). The mTEM processes and encodes these signals into a multidimensional spike train using a population of $N$ neurons. For each neuron $i=1,...,N$, the results of this processing are aggregated into the dendritic current $v^i$ flowing into the spike initiation zone, where it is encoded into a time sequence $(t_k^i)_{k\in\mathbb{Z}}$, with $t_k^i$ denoting the timing of the $k$th spike of neuron $i$.

Similarly to traditional TEMs, mTEMs can employ a myriad of spiking neuron models. Several examples include conductance-based models such as Hodgkin-Huxley, Morris-Lecar, Fitzhugh-Nagumo, Wang-Buzsaki, Hindmarsh-Rose [20] as well as simpler models such as the ideal and leaky integrate-and-fire (IAF) neurons [15]. For clarity, we will limit our discussion to the ideal IAF neuron, since other models can be handled as described previously [20, 21]. For an ideal IAF neuron with a bias $b^i \in \mathbb{R}_+$, capacitance $C^i \in \mathbb{R}_+$ and threshold $\delta^i \in \mathbb{R}_+$ (Fig. 1b), the mapping of the current $v^i$ into spikes is described by a set of equations formerly known as the t-transform [18]:

$$\int_{t_k^i}^{t_{k+1}^i} v^i(s)ds = q_k^i, \qquad k \in \mathbb{Z}, \tag{1}$$

where $q_k^i = C^i\delta^i - b^i(t_{k+1}^i - t_k^i)$. Intuitively, at every spike time $t_{k+1}^i$, the ideal IAF neuron is providing a measurement $q_k^i$ of the current $v^i(t)$ on the time interval $[t_k^i, t_{k+1}^i]$.

## 2.1 Modeling Sensory Inputs

We model input signals as elements of reproducing kernel Hilbert spaces (RKHSs) [22]. Most real world signals, including natural stimuli can be described by an appropriately chosen RKHS [23]. For practical and computational reasons we choose to work with the space of trigonometric polynomials $\mathcal{H}_{n_m}$ defined below, where each element of the space is a function in $n_m$ variables ($n_m \in \mathbb{N}, m = 1, 2, ..., M$). However, we note that the results obtained in this paper are not limited to this particular choice of RKHS (see, e.g., [24]).

**Definition 1.** *The space of trigonometric polynomials $\mathcal{H}_{n_m}$ is a Hilbert space of complex-valued functions*

$$u_{n_m}^m (x_1, ..., x_{n_m}) = \sum_{l_1=-L_1}^{L_1} \cdots \sum_{l_{n_m}=-L_{n_m}}^{L_{n_m}} u_{l_1...l_{n_m}}^m e_{l_1...l_{n_m}} (x_1, ..., x_{n_m}),$$

*over the domain $\mathbb{D}_{n_m} = \prod_{n=1}^{n_m} [0, T_n]$, where $u_{l_1...l_{n_m}}^m \in \mathbb{C}$ and the functions $e_{l_1...l_{n_m}} (x_1, ..., x_{n_m}) = \exp\left(\sum_{n=1}^{n_m} jl_n\Omega_n x_n/L_n\right)/\sqrt{T_1 \cdots T_{n_m}}$, with $j$ denoting the imaginary number. Here $\Omega_n$ is the bandwidth, $L_n$ is the order, and $T_n = 2\pi L_n/\Omega_n$ is the period in dimension $x_n$. $\mathcal{H}_{n_m}$ is endowed with the inner product $\langle \cdot, \cdot \rangle : \mathcal{H}_{n_m} \times \mathcal{H}_{n_m} \to \mathbb{C}$, where*

$$\langle u_{n_m}^m, w_{n_m}^m \rangle = \int_{\mathbb{D}_{n_m}} u_{n_m}^m (x_1, ..., x_{n_m}) \overline{w_{n_m}^m (x_1, ..., x_{n_m})} dx_1...dx_{n_m}. \tag{2}$$

*Given the inner product in (2), the set of elements $e_{l_1...l_{n_m}} (x_1, ..., x_{n_m})$ forms an orthonormal basis in $\mathcal{H}_{n_m}$. Moreover, $\mathcal{H}_{n_m}$ is an RKHS with the reproducing kernel (RK)*

$$K_{n_m} (x_1, ..., x_{n_m}; y_1, ..., y_{n_m}) = \sum_{l_1=-L_1}^{L_1} \cdots \sum_{l_{n_m}=-L_{n_m}}^{L_{n_m}} e_{l_1...l_{n_m}} (x_1, ..., x_{n_m}) \overline{e_{l_1...l_{n_m}} (y_1, ..., y_{n_m})}.$$

**Remark 1.** *In what follows, we will primarily be concerned with time-varying stimuli, and the dimension $x_{n_m}$ will denote the temporal dimension t of the stimulus $u_{n_m}^m$, i.e., $x_{n_m} = t$.*

**Remark 2.** *For $M$ concurrently received stimuli, we have $T_{n_1} = T_{n_2} = \cdots = T_{n_M}$.*

**Example 1.** *We model audio stimuli $u_1^m = u_1^m(t)$ as elements of the RKHS $\mathcal{H}_1$ over the domain $\mathbb{D}_1 = [0, T_1]$. For notational convenience, we drop the dimensionality subscript and use $T$, $\Omega$ and $L$, to denote the period, bandwidth and order of the space $\mathcal{H}_1$. An audio signal $u_1^m \in \mathcal{H}_1$ can be written as $u_1^m(t) = \sum_{l=-L}^{L} u_l^m e_l(t)$, where the coefficients $u_l^m \in \mathbb{C}$ and $e_l(t) = \exp\left(jl\Omega t/L\right)/\sqrt{T}$.*

**Example 2.** *We model video stimuli $u_3^m = u_3^m(x, y, t)$ as elements of the RKHS $\mathcal{H}_3$ defined on $\mathbb{D}_3 = [0, T_1] \times [0, T_2] \times [0, T_3]$, where $T_1 = 2\pi L_1/\Omega_1$, $T_2 = 2\pi L_2/\Omega_2$, $T_3 = 2\pi L_3/\Omega_3$, with $(\Omega_1, L_1)$, $(\Omega_2, L_2)$ and $(\Omega_3, L_3)$ denoting the (bandwidth, order) pairs in spatial directions $x$, $y$ and in time $t$, respectively. A video signal $u_3^m \in \mathcal{H}_3$ can be written as $u_3^m(x, y, t) = \sum_{l_1=-L_1}^{L_1} \sum_{l_2=-L_2}^{L_2} \sum_{l_3=-L_3}^{L_3} u_{l_1l_2l_3}^m e_{l_1l_2l_3}(x, y, t)$, where the coefficients $u_{l_1l_2l_3}^m \in \mathbb{C}$ and the functions $e_{l_1l_2l_3}(x, y, t) = \exp\left(jl_1\Omega_1 x/L_1 + jl_2\Omega_2 y/L_2 + jl_3\Omega_3 t/L_3\right)/\sqrt{T_1T_2T_3}$.*

## 2.2 Modeling Sensory Processing

Multisensory processing can be described by a nonlinear dynamical system capable of modeling linear and nonlinear stimulus transformations, including cross-talk between stimuli [25]. For clarity, here we will consider only the case of linear transformations that can be described by a linear filter having an impulse response, or kernel, $h_{n_m}^m(x_1, ..., x_{n_m})$. The kernel is assumed to be bounded-input bounded-output (BIBO)-stable and causal. Without loss of generality, we assume that such transformations involve convolution in the time domain (temporal dimension $x_{n_m}$) and integration in dimensions $x_1, ..., x_{n_m-1}$. We also assume that the kernel has a finite support in each direction $x_n, n = 1, ..., n_m$. In other words, the kernel $h_{n_m}^m$ belongs to the space $H_{n_m}$ defined below.

**Definition 2.** *The filter kernel space $H_{n_m} = \left\{ h_{n_m}^m \in \mathbb{L}^1(\mathbb{R}^{n_m}) \mid \text{supp}(h_{n_m}^m) \subseteq \mathbb{D}_{n_m} \right\}$.*

**Definition 3.** *The projection operator $\mathcal{P} : H_{n_m} \to \mathcal{H}_{n_m}$ is given (by abuse of notation) by*

$$(\mathcal{P}h_{n_m}^m)(x_1, ..., x_{n_m}) = \langle h_{n_m}^m(\cdot, ..., \cdot), K_{n_m}(\cdot, ..., \cdot; x_1, ..., x_{n_m}) \rangle. \tag{3}$$

*Since $\mathcal{P}h_{n_m}^m \in \mathcal{H}_{n_m}$, $(\mathcal{P}h_{n_m}^m)(x_1, ..., x_{n_m}) = \sum_{l_1=-L_1}^{L_1} \cdots \sum_{l_{n_m}=-L_{n_m}}^{L_{n_m}} h_{l_1...l_{n_m}}^m e_{l_1...l_{n_m}}(x_1, ..., x_{n_m})$.*

# 3 Multisensory Decoding

Consider an mTEM comprised of a population of $N$ ideal IAF neurons receiving $M$ input signals $u_{n_m}^m$ of dimensions $n_m$, $m = 1, ..., M$. Assuming that the multisensory processing is given by kernels $h_{n_m}^{im}$, $m = 1, ..., M$, $i = 1, ..., N$, the t-transform in (1) can be rewritten as

$$\mathcal{T}_k^{i1}[u_{n_1}^1] + \mathcal{T}_k^{i2}[u_{n_2}^2] + ... + \mathcal{T}_k^{iM}[u_{n_M}^M] = q_k^i, \qquad k \in \mathbb{Z}, \tag{4}$$

where $\mathcal{T}_k^{im} : \mathcal{H}_{n_m} \to \mathbb{R}$ are linear functionals defined by

$$\mathcal{T}_k^{im}[u_{n_m}^m] = \int_{t_k^i}^{t_{k+1}^i} \left[ \int_{\mathbb{D}_{n_m}} h_{n_m}^{im}(x_1, ..., x_{n_m-1}, s) u_{n_m}^m(x_1, ..., x_{n_m-1}, t-s) dx_1 ... dx_{n_m-1} ds \right] dt.$$

We observe that each $q_k^i$ in (4) is a real number representing a quantal measurement of all $M$ stimuli, taken by the neuron $i$ on the interval $[t_k^i, t_{k+1}^i)$. These measurements are produced in an asynchronous fashion and can be computed directly from spike times $(t_k^i)_{k \in \mathbb{Z}}$ using (1). We now demonstrate that it is possible to reconstruct stimuli $u_{n_m}^m$, $m = 1, ..., M$ from $(t_k^i)_{k \in \mathbb{Z}}$, $i = 1, ..., N$.

**Theorem 1.** *(Multisensory Time Decoding Machine (mTDM))*
*Let $M$ signals $u_{n_m}^m \in \mathcal{H}_{n_m}$ be encoded by a multisensory TEM comprised of $N$ ideal IAF neurons and $N \times M$ receptive fields with full spectral support. Assume that the IAF neurons do not have the same parameters, and/or the receptive fields for each modality are linearly independent. Then given the filter kernel coefficients $h_{l_1 ... l_{n_m}}^{im}$, $i = 1, ..., N$, all inputs $u_{n_m}^m$ can be perfectly recovered as*

$$u_{n_m}^m(x_1, ..., x_{n_m}) = \sum_{l_1=-L_1}^{L_1} ... \sum_{l_{n_m}=-L_{n_m}}^{L_{n_m}} u_{l_1 ... l_{n_m}}^m e_{l_1 ... l_{n_m}}(x_1, ..., x_{n_m}), \tag{5}$$

*where $u_{l_1 ... l_{n_m}}^m$ are elements of $\mathbf{u} = \mathbf{\Phi}^+ \mathbf{q}$, and $\mathbf{\Phi}^+$ denotes the pseudoinverse of $\mathbf{\Phi}$. Furthermore, $\mathbf{\Phi} = [\mathbf{\Phi}^1; \mathbf{\Phi}^2; ...; \mathbf{\Phi}^N]$, $\mathbf{q} = [\mathbf{q}^1; \mathbf{q}^2; ...; \mathbf{q}^N]$ and $[\mathbf{q}^i]_k = q_k^i$. Each matrix $\mathbf{\Phi}^i = [\mathbf{\Phi}^{i1}, \mathbf{\Phi}^{i2}, ..., \mathbf{\Phi}^{im}]$, with*

$$[\mathbf{\Phi}^{im}]_{kl} = \begin{cases} h_{-l_1, -l_2, ..., -l_{n_m-1}, l_{n_m}}^{im}(t_{k+1}^i - t_k^i), & l_{n_m} = 0 \\ h_{-l_1, -l_2, ..., -l_{n_m-1}, l_{n_m}}^{im} \dfrac{L_{n_m} \sqrt{T_{n_m}} \left( e_{l_{n_m}}(t_{k+1}^i) - e_{l_{n_m}}(t_k^i) \right)}{j l_{n_m} \Omega_{n_m}}, & l_{n_m} \neq 0 \end{cases}, \tag{6}$$

*where the column index $l$ traverses all possible subscript combinations of $l_1, l_2, ..., l_{n_m}$. A necessary condition for recovery is that the total number of spikes generated by all neurons is larger than $\sum_{m=1}^M \prod_{n=1}^{n_m}(2L_n+1) + N$. If each neuron produces $\nu$ spikes in an interval of length $T_{n_1} = T_{n_2} = \cdots = T_{n_M}$, a sufficient condition is $N \geq \left\lceil \sum_{m=1}^M \prod_{n=1}^{n_m}(2L_n+1) / \min(\nu - 1, 2L_{n_m}+1) \right\rceil$, where $\lceil x \rceil$ denotes the smallest integer greater than $x$.*

**Proof:** Substituting (5) into (4), $q_k^i = \mathcal{T}_k^{i1}[u_{n_1}^1] + ... + \mathcal{T}_k^{iM}[u_{n_M}^M] = \langle u_{n_1}^1, \phi_{1k}^{i1} \rangle + ... + \langle u_{n_M}^M, \phi_{Mk}^{iM} \rangle = \sum_{l_1} ... \sum_{l_{n_1}} u_{-l_1, -l_2, -l_{n_1-1}, l_{n_1}}^1 \overline{\phi_{l_1 ... l_{n_1} k}^{i1}} + ... + \sum_{l_1} ... \sum_{l_{n_M}} u_{-l_1, -l_2, -l_{n_M-1}, l_{n_M}}^M \overline{\phi_{l_1 ... l_{n_M} k}^{iM}}$, where $k \in \mathbb{Z}$ and the second equality follows from the Riesz representation theorem with $\phi_{n_m k}^{im} \in \mathcal{H}_{n_m}$, $m = 1, ..., M$. In matrix form the above equality can be written as $\mathbf{q}^i = \mathbf{\Phi}^i \mathbf{u}$, with $[\mathbf{q}^i]_k = q_k^i$, $\mathbf{\Phi}^i = [\mathbf{\Phi}^{i1}, \mathbf{\Phi}^{i2}, ..., \mathbf{\Phi}^{iM}]$, where elements $[\mathbf{\Phi}^{im}]_{kl}$ are given by $[\mathbf{\Phi}^{im}]_{kl} = \overline{\phi_{l_1 ... l_{n_m} k}^{im}}$, with index $l$ traversing all possible subscript combinations of $l_1, l_2, ..., l_{n_m}$. To find the coefficients $\overline{\phi_{l_1 ... l_{n_m} k}^{im}}$, we note that $\phi_{l_1 ... l_{n_m} k}^{im} = \overline{\mathcal{T}_{n_m k}^{im}(e_{l_1 ... l_{n_m}})}$, $m = 1, ..., M$, $i = 1, ..., N$. The column vector $\mathbf{u} = [\mathbf{u}^1; \mathbf{u}^2; ...; \mathbf{u}^m]$ with the vector $\mathbf{u}^m$ containing $\prod_{n=1}^{n_m}(2L_n+1)$ entries corresponding to coefficients $u_{l_1 l_2 ... l_{n_m}}^m$. Repeating for all neurons $i = 1, ..., N$, we obtain $\mathbf{q} = \mathbf{\Phi} \mathbf{u}$ with $\mathbf{\Phi} = [\mathbf{\Phi}^1; \mathbf{\Phi}^2; ...; \mathbf{\Phi}^N]$ and $\mathbf{q} = [\mathbf{q}^1; \mathbf{q}^2; ...; \mathbf{q}^N]$. This system of linear equations can be solved for $\mathbf{u}$, provided that the rank $r(\mathbf{\Phi})$ of matrix $\mathbf{\Phi}$ satisfies $r(\mathbf{\Phi}) = \sum_m \prod_{n=1}^{n_m}(2L_n+1)$. A necessary condition for the latter is that the total number of measurements generated by all $N$ neurons is greater or equal to $\prod_{n=1}^{n_m}(2L_n+1)$. Equivalently, the total number of spikes produced by all $N$ neurons should be greater than $\prod_{n=1}^{n_m}(2L_n+1) + N$. Then $\mathbf{u}$ can be uniquely specified as the solution to a convex optimization problem, e.g., $\mathbf{u} = \mathbf{\Phi}^+ \mathbf{q}$. To find the sufficient condition, we note

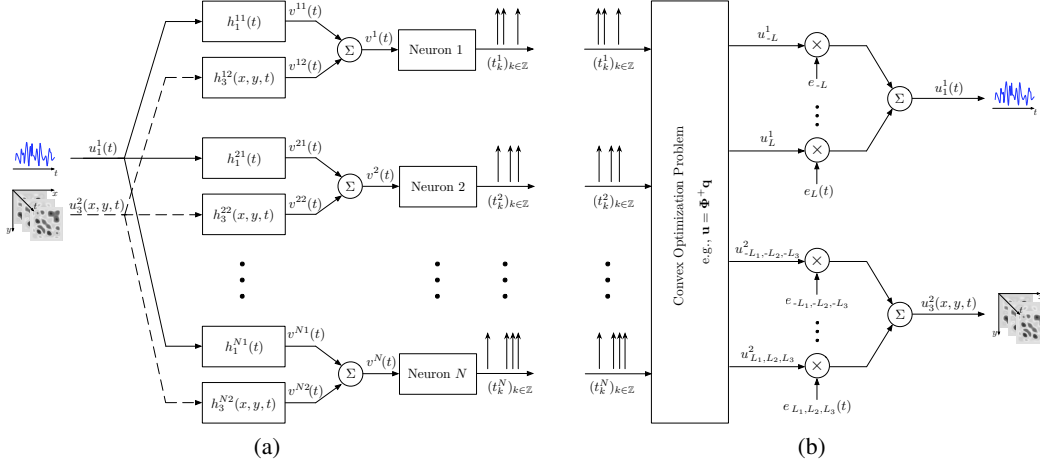

Figure 2: **Multimodal TEM & TDM for audio and video integration (a)** Block diagram of the multimodal TEM. **(b)** Block diagram of the multimodal TDM.

that the $m^{\text{th}}$ component $v^{im}$ of the dendritic current $v^i$ has a maximal bandwidth of $\Omega_{n_m}$ and we need only $2L_{n_m} + 1$ measurements to specify it. Thus each neuron can produce a maximum of only $2PL_{n_m} + 1$ informative measurements, or equivalently, $2PL_{n_m} + 2$ informative spikes on a time interval $[0, T_{n_m}]$. It follows that for each modality, we require at least $\prod_{n=1}^{n_m}(2L_n + 1)/(2L_{n_m} + 1)$ neurons if $\nu \geq (2L_{n_m} + 2)$ and at least $\lceil \prod_{n=1}^{n_m}(2L_n + 1)/(\nu - 1) \rceil$ neurons if $\nu < (2L_{n_m} + 2)$. $\square$

## 4 Multisensory Identification

We now investigate the following *nonlinear* neural identification problem: given stimuli $u_{n_m}^m$, $m = 1, ..., M$, at the input to a multisensory neuron $i$ and spikes at its output, find the multisensory receptive field kernels $h_{n_m}^{im}$, $m = 1, ..., M$. We will show that this problem is mathematically dual to the decoding problem discussed above. Specifically, we will demonstrate that the identification problem can be converted into a neural encoding problem, where each spike train $(t_k^i)_{k \in \mathbb{Z}}$ produced during an experimental trial $i$, $i = 1, ..., N$, is interpreted to be generated by the $i^{\text{th}}$ neuron in a population of $N$ neurons. We consider identifying kernels for only one multisensory neuron (identification for multiple neurons can be performed in a serial fashion) and drop the superscript $i$ in $h_{n_m}^{im}$ throughout this section. Instead, we introduce the natural notion of performing multiple experimental trials and use the same superscript $i$ to index stimuli $u_{n_m}^{im}$ on different trials $i = 1, ..., N$.

Consider the multisensory neuron depicted in Fig. 1. Since for every trial $i$, an input signal $u_{n_m}^{im}$, $m = 1, ..., M$, can be modeled as an element of some space $\mathcal{H}_{n_m}$, we have $u_{n_m}^{im}(x_1, ..., x_{n_m}) = \langle u_{n_m}^{im}(\cdot, ..., \cdot), K_{n_m}(\cdot, ..., \cdot; x_1, ..., x_{n_m}) \rangle$ by the reproducing property of the RK $K_{n_m}$. It follows that

$$\int_{\mathbb{D}_{n_m}} h_{n_m}^m(s_1, ..., s_{n_m-1}, s_{n_m}) u_{n_m}^{im}(s_1, ..., s_{n_m-1}, t - s_{n_m}) ds_1...ds_{n_m-1}ds_{n_m} =$$

$$\overset{(a)}{=} \int_{\mathbb{D}_{n_m}} u_{n_m}^{im}(s_1, ..., s_{n_m-1}, s_{n_m}) \left\langle h_{n_m}^m(\cdot, ..., \cdot), K_{n_m}(\cdot, ..., \cdot; s_1, ..., s_{n_m-1}, t - s_{n_m}) \right\rangle ds_1...ds_{n_m} =$$

$$\overset{(b)}{=} \int_{\mathbb{D}_{n_m}} u_{n_m}^{im}(s_1, ..., s_{n_m-1}, s_{n_m}) (\mathcal{P}h_{n_m}^m)(s_1, ..., s_{n_m-1}, t - s_{n_m}) ds_1...ds_{n_m-1}ds_{n_m},$$

where $^{(a)}$ follows from the reproducing property and symmetry of $K_{n_m}$ and Definition 2, and $^{(b)}$ from the definition of $\mathcal{P}h_{n_m}^m$ in (3). The t-transform of the mTEM in Fig. 1 can then be written as

$$\mathcal{L}_k^{i1}[\mathcal{P}h_{n_1}^1] + \mathcal{L}_k^{i2}[\mathcal{P}h_{n_2}^2] + ... + \mathcal{L}_k^{iM}[\mathcal{P}h_{n_M}^M] = q_k^i, \tag{7}$$

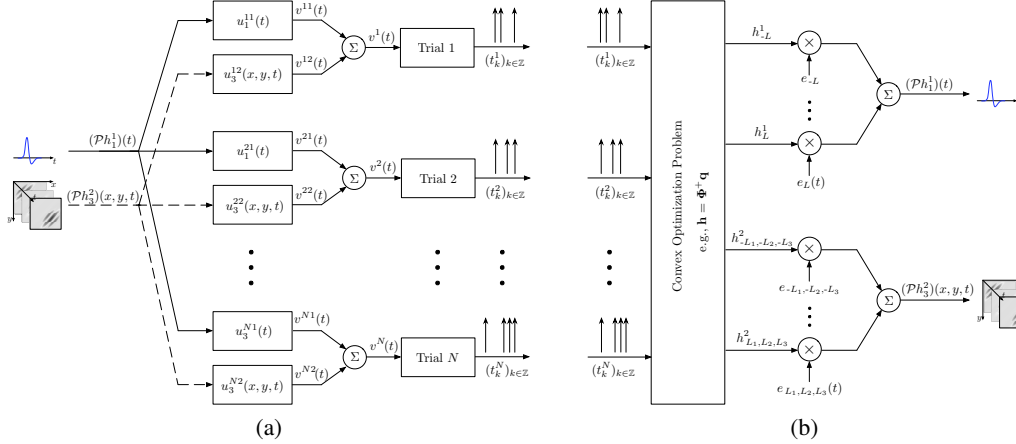

(a)                                                                    (b)

Figure 3: **Multimodal CIM for audio and video integration (a)** Time encoding interpretation of the multimodal CIM. **(b)** Block diagram of the multimodal CIM.

where $\mathcal{L}_k^{im} : \mathcal{H}_{n_m} \to \mathbb{R}$, $m = 1, ..., M$, $k \in \mathbb{Z}$, are linear functionals defined by

$$\mathcal{L}_k^{im}[\mathcal{P}h_{n_m}^m] = \int_{t_k^i}^{t_{k+1}^i} \left[ \int_{\mathbb{D}_m} u_{n_m}^{im}(s_1, \, ..., \, s_{n_m})(\mathcal{P}h_{n_m}^m)(s_1, ..., t - s_{n_m}) ds_1 \, ... \, ds_{n_m} \right] dt.$$

**Remark 3.** *Intuitively, each inter-spike interval $[t_k^i, t_{k+1}^i)$ produced by the IAF neuron is a time measurement $q_k^i$ of the (weighted) sum of all kernel projections $\mathcal{P}h_{n_m}^m$, $m = 1, ..., M$.*

**Remark 4.** *Each projection $\mathcal{P}h_{n_m}^m$ is determined by the corresponding stimuli $u_{n_m}^{im}$, $i = 1, ..., N$, employed during identification and can be substantially different from the underlying kernel $h_{n_m}^m$.*

It follows that we should be able to identify the projections $\mathcal{P}h_{n_m}^m$, $m = 1, ..., M$, from the measurements $(q_k^i)_{k \in \mathbb{Z}}$. Since we are free to choose any of the spaces $\mathcal{H}_{n_m}$, an arbitrarily-close identification of original kernels is possible, provided that the bandwidth of the test signals is sufficiently large.

**Theorem 2.** *(Multisensory Channel Identification Machine (mCIM))*
*Let $\{\mathbf{u}^i\}_{i=1}^N$, $\mathbf{u}^i = [u_{n_1}^{i1}, ..., u_{n_M}^{iM}]^T$, $u_{n_m}^{im} \in \mathcal{H}_{n_m}$, $m = 1, ..., M$, be a collection of $N$ linearly independent stimuli at the input to an mTEM circuit comprised of receptive fields with kernels $h_{n_m}^m \in H_{n_m}$, $m = 1, ..., M$, in cascade with an ideal IAF neuron. Given the coefficients $u_{l_1,...,l_{n_m}}^{im}$ of stimuli $u_{n_m}^{im}$, $i = 1, ..., N$, $m = 1, ..., M$, the kernel projections $\mathcal{P}h_{n_m}^m$, $m = 1, ..., M$, can be perfectly identified as $(\mathcal{P}h_{n_m}^m)(x_1, ..., x_{n_m}) = \sum_{l_1=-L_1}^{L_1} ... \sum_{l_{n_m}=-L_{n_m}}^{L_{n_m}} h_{l_1...l_{n_m}}^m e_{l_1...l_{n_m}}(x_1, ..., x_{n_m})$, where $h_{l_1...l_{n_m}}^m$ are elements of $\mathbf{h} = \mathbf{\Phi}^+ \mathbf{q}$, and $\Phi^+$ denotes the pseudoinverse of $\Phi$. Furthermore, $\mathbf{\Phi} = [\mathbf{\Phi}^1; \, \mathbf{\Phi}^2; \, ...; \mathbf{\Phi}^N]$, $\mathbf{q} = [\mathbf{q}^1; \, \mathbf{q}^2; \, ...; \mathbf{q}^N]$ and $[\mathbf{q}^i]_k = q_k^i$. Each matrix $\mathbf{\Phi}^i = [\mathbf{\Phi}^{i1}, \mathbf{\Phi}^{i2}, ..., \mathbf{\Phi}^{im}]$, with*

$$[\mathbf{\Phi}^{im}]_{kl} = \begin{cases} u_{-l_1,-l_2,...,-l_{n_m-1},l_{n_m}}^{im} (t_{k+1}^i - t_k^i), & l_{n_m} = 0 \\ \dfrac{u_{-l_1,-l_2,...,-l_{n_m-1},l_{n_m}}^{im} L_{n_m} \sqrt{T_{n_m}} \left( e_{l_{n_m}}(t_{k+1}^i) - e_{l_{n_m}}(t_k^i) \right)}{j l_{n_m} \Omega_{n_m}}, & l_{n_m} \neq 0 \end{cases}, \quad (8)$$

*where $l$ traverses all subscript combinations of $l_1, l_2, ..., l_{n_m}$. A necessary condition for identification is that the total number of spikes generated in response to all $N$ trials is larger than $\sum_{m=1}^M \prod_{n=1}^{n_m} (2L_n + 1) + N$. If the neuron produces $\nu$ spikes on each trial, a sufficient condition is that the number of trials $N \geq \left\lceil \sum_{m=1}^M \prod_{n=1}^{n_m} (2L_n + 1) / \min(\nu - 1, 2L_{n_m} + 1) \right\rceil$.*

**Proof:** The equivalent representation of the t-transform in equations (4) and (7) implies that the decoding of the stimulus $u_{n_m}^m$ (in Theorem 1) and the identification of the filter projections $\mathcal{P}h_{n_m}^m$ encountered here are dual problems. Therefore, the receptive field identification problem is equivalent to a neural encoding problem: the projections $\mathcal{P}h_{n_m}^m$, $m = 1, ..., M$, are encoded with an mTEM comprised of $N$ neurons and receptive fields $u_{n_m}^{im}$, $i = 1, ..., N$, $m = 1, ..., M$. The algorithm for finding the coefficients $h_{l_1...l_{n_m}}^m$ is analogous to the one for $u_{l_1...l_{n_m}}^m$ in Theorem 1.

# 5 Examples

A simple (mono) audio/video TEM realized using a bank of temporal and spatiotemporal linear filters and a population of integrate-and-fire neurons, is shown in Fig. 2. An analog audio signal $u_1^1(t)$ and an analog video signal $u_2^2(x, y, t)$ appear as inputs to temporal filters with kernels $h_1^{i1}(t)$ and spatiotemporal filters with kernels $h_3^{i2}(x, y, t)$, $i = 1, ..., N$. Each temporal and spatiotemporal filter could be realized in a number of ways, e.g., using gammatone and Gabor filter banks. For simplicity, we assume that the number of temporal and spatiotemporal filters in Fig. 2 is the same. In practice, the number of components could be different and would be determined by the bandwidth of input stimuli $\Omega$, or equivalently the order $L$, and the number of spikes produced (Theorems 1-2).

For each neuron $i$, $i = 1, ..., N$, the filter outputs $v^{i1}$ and $v^{i2}$, are summed to form the aggregate dendritic current $v^i$, which is encoded into a sequence of spike times $(t_k^i)_{k \in \mathbb{Z}}$ by the $i^{\text{th}}$ integrate-and-fire neuron. Thus each spike train $(t_k^i)_{k \in \mathbb{Z}}$ carries information about two stimuli of completely different modalities (audio and video) and, under certain conditions, the entire collection of spike trains $\{t_k^i\}_{i=1}^N$, $k \in \mathbb{Z}$, can provide a faithful representation of both signals.

To demonstrate the performance of the algorithm presented in Theorem 1, we simulated a multisensory TEM with each neuron having a non-separable spatiotemporal receptive field for video stimuli and a temporal receptive field for audio stimuli. Spatiotemporal receptive fields were chosen randomly and had a bandwidth of $4\,\text{Hz}$ in temporal direction $t$ and $2\,\text{Hz}$ in each spatial direction $x$ and $y$. Similarly, temporal receptive fields were chosen randomly from functions bandlimited to $4\,\text{kHz}$. Thus, two distinct stimuli having different dimensions (three for video, one for audio) and dynamics (2-4 cycles vs. $4,000$ cycles in each direction) were multiplexed at the level of every spiking neuron and encoded into an unlabeled set of spikes. The mTEM produced a total of $360,000$ spikes in response to a 6-second-long grayscale video and mono audio of Albert Einstein explaining the mass-energy equivalence formula $E = mc^2$: "... [a] very small amount of mass may be converted into a very large amount of energy." A multisensory TDM was then used to reconstruct the video and audio stimuli from the produced set of spikes. Fig. 4a-b shows the original (top row) and recovered (middle row) video and audio, respectively, together with the error between them (bottom row).

The neural encoding interpretation of the identification problem for the grayscale video/mono audio TEM is shown in Fig. 3a. The block diagram of the corresponding mCIM appears in Fig. 3b. Comparing this diagram to the one in Fig. 2, we note that neuron blocks have been replaced by trial blocks. Furthermore, the stimuli now appear as kernels describing the filters and the inputs to the circuit are kernel projections $\mathcal{P}h_{n_m}^m$, $m = 1, ..., M$. In other words, identification of a *single* neuron has been converted into a *population* encoding problem, where the artificially constructed population of $N$ neurons is associated with the $N$ spike trains generated in response to $N$ experimental trials.

The performance of the mCIM algorithm is visualized in Fig. 5. Fig. 5a-b shows the original (top row) and recovered (middle row) spatio-temporal and temporal receptive fields, respectively, together with the error between them (bottom row).

# 6 Conclusion

We presented a spiking neural circuit for multisensory integration that encodes multiple information streams, e.g., audio and video, into a single spike train at the level of individual neurons. We derived conditions for inverting the nonlinear operator describing the multiplexing and encoding in the spike domain and developed methods for identifying multisensory processing using concurrent stimulus presentations. We provided explicit algorithms for multisensory decoding and identification and evaluated their performance using natural audio and video stimuli. Our investigations brought to light a key duality between identification of multisensory processing in a single neuron and the recovery of stimuli encoded with a population of multisensory neurons. Given the powerful machinery of employed RKHSs, extensions to neural circuits with noisy neurons are straightforward [15, 23].

## Acknowledgement

The work presented here was supported in part by AFOSR under grant #FA9550-12-1-0232 and, in part, by NIH under the grant #R021 DC012440001.

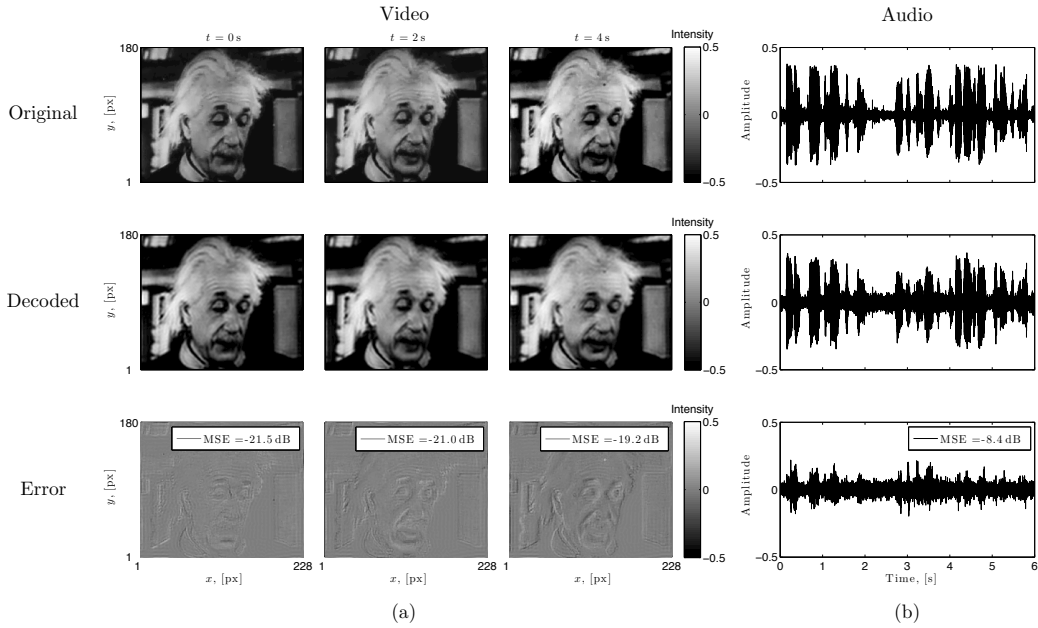

(a)  (b)

Figure 4: **Multisensory decoding. (a)** Grayscale Video Recovery. (top row) Three frames of the original grayscale video $u_3^2$. (middle row) Corresponding three frames of the decoded video projection $\mathcal{P}_3 u_3^2$. (bottom row) Error between three frames of the original and identified video. $\Omega_1 = 2\pi \cdot 2\,\mathrm{rad/s}$, $L_1 = 30$, $\Omega_2 = 2\pi \cdot 36/19\,\mathrm{rad/s}$, $L_2 = 36$, $\Omega_3 = 2\pi \cdot 4\,\mathrm{rad/s}$, $L_3 = 4$. **(b)** Mono Audio Recovery. (top row) Original mono audio signal $u_1^1$. (middle row) Decoded projection $\mathcal{P}_1 u_1^1$. (bottom row) Error between the original and decoded audio. $\Omega = 2\pi \cdot 4,000\,\mathrm{rad/s}$, $L = 4,000$. **Click here to see and hear the decoded video and audio stimuli.**

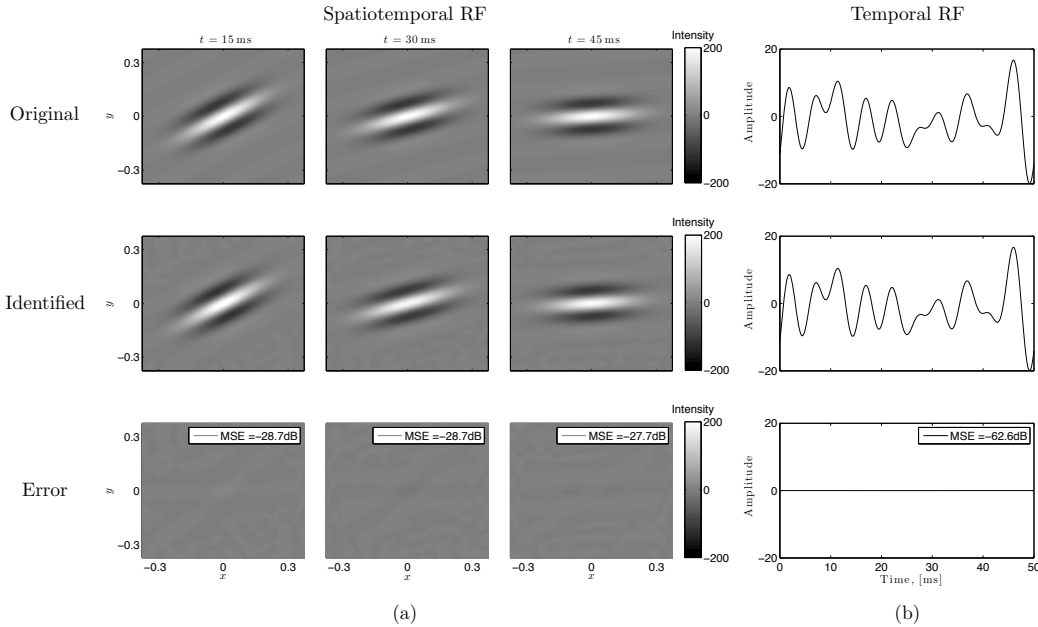

(a)  (b)

Figure 5: **Multisensory identification. (a)** (top row) Three frames of the original spatiotemporal kernel $h_3^2(x, y, t)$. Here, $h_3^2$ is a spatial Gabor function rotating clockwise in space as a function of time. (middle row) Corresponding three frames of the identified kernel $\mathcal{P}h_3^{2*}(x, y, t)$. (bottom row) Error between three frames of the original and identified kernel. $\Omega_1 = 2\pi \cdot 12\,\mathrm{rad/s}$, $L_1 = 9$, $\Omega_2 = 2\pi \cdot 12\,\mathrm{rad/s}$, $L_2 = 9$, $\Omega_3 = 2\pi \cdot 100\,\mathrm{rad/s}$, $L_3 = 5$. **(b)** Identification of the temporal RF (top row) Original temporal kernel $h_1^1(t)$. (middle row) Identified projection $\mathcal{P}h_1^{1*}(t)$. (bottom row) Error between $h_1^1$ and $\mathcal{P}h_1^{1*}$. $\Omega = 2\pi \cdot 200\,\mathrm{rad/s}$, $L = 10$.

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
