[Reviews · NeurIPS 2013]

Submitted by Assigned_Reviewer_6

The present paper continues a line of successful previous papers dealing with the encoding of signals through the so-called TEM (time encoding machine). The present paper is motivated by the recent realization within the neuroscience community that most sensory processing is in multi-modal, and that this occurs already at early stages of the processing hierarchy. The signal processing advantages of such a scheme are intuitively obvious, and the present paper contributes to demonstrating a practical and mathematically well-founded approach. Such an approach offers the potential advantage of natural hardware implementation using low power elements.
The mathematical framework for signal representation is based on reproducing kernel Hilbert spaces, specifically focusing here on trigonometric functions. The main mathematical contributions appear in the form of two theorems, which establish, respectively, the decoding property and the equivalence of filter identification and neural encoding. Several examples are also presented which demonstrate the effectiveness and power of the method.
This paper deals with an important and relevant problem in computational neuroscience (and signal processing) and suggests a solid and effective solution. Although the method proposed does not seem to be biologically plausible form an algorithmic perspective, it does suggest limits to what can be achieved with the setting delineated. I was missing some discussion of robustness to noise and power consumption considerations, which was one of the motivations for the paper.
Summary: An extension of previous work providing a framework for encoding and decoding multi-sensory continuous signals in the spike domain. Precise mathematical characterizations and numerical demonstrations suggest the method is well founded both mathematically and algorithmically.

Submitted by Assigned_Reviewer_7

Reviewer's response to the rebuttal:



'"Theorem 1 needs to be corrected/clarified."
We apologize for the confusion: part of the theorem was accidentally omitted. The sufficient condition is to have a large enough population of (different) neurons: N >= SUM_m[(2L_1+1)(2L_2+1)…(2L_{n_m-1} + 1)]. The necessary condition is that the total number of spikes is greater than
(2L_{n_m}+1)*SUM_m[(2L_1+1)(2L_2+1)…(2L_{n_m-1} + 1)] +N"

OK, that seems more acceptable, even though I don't fully understand the condition -- what prevents these spikes from being completely synchronized across neurons, and hence being uninformative?

'"…what would happen if all filter kernels were the same? Then no matter how many neurons you added, you would not gain any further information.”
That is not entirely correct. For example, if all filters are Dirac-delta functions, the inputs are encoded directly without processing. However, one can recover all inputs if the neurons have different parameters (e.g., bias, capacitance, threshold of the IAF neuron). Imposing linear independence between kernels is another way to guarantee signal recovery.'

True, but in your definition of the IAF-TEM (eqn3 and following) it does not appear that the neurons have different parameters, hence my remark. Please say that somewhere.


Review for "Multisensory Encoding, Decoding, and Identification"

The paper describes an extension of Time Encoding Machines (TEM) to the multiple-input-multiple-output (MIMO) setting, using the formalism of Reproducing Kernel Hilbert Spaces (RHKS). The approach is motivated from a neuroscientific perspective: in the introduction, it is claimed (correctly) that multisensory integration (and processing) provides many benefits to organisms, but that it is still poorly understood. What follows is a particular extension of TEMs
to the MIMO setting, with trignometric polynomials as basis functions. Necessary conditions for perfect decoding and/or system identification are given. The feasibility of the resulting algorithms are demonstrated on an audio/video decoding task, and an audio/video encoder identification task. The paper concludes with the remark that this work constitutes the first tractable computational model for multisensory integration (which is wrong) and that extensions to the noisy case would be straightforward.

Clarity:
The paper is mostly well written, even though the notation in section 3 is cumbersome, but I do not see how that could be avoided.

Originality: the paper's most original contribution is the extensions of TEMs to the MIMO setting, the connection to the brain is tenuous.

Quality: I believe some parts of the paper need to be improved before publication.

Significance: from a technical perspective, the paper demonstrates the feasibility of TEMs for multi-modal encoding, which might prove useful.
There is no clear contribution to neural coding, other than the motivation in the introduction.


Detailed comments:

line 100: "bandlmited" -> bandlimited

line 137: j is the complex unit, please say that somewhere.

line 144: the u_n in eqn.(1) are not the same as the u_n^i in figure 1, right? I propose to use a different letter.

line 166: "BIBO-stable" bounded-input-bounded-output? please define.

line 166: "...finite temporal support of lenghth S_i \leq T_i...": would that be spatio-temporal support for the video filters in example 2?

line 177: Say here that the h_{l_1 \ldots l_n} are the filter coefficients, to connect this formula to theorem 1.

line 182: "...IAF TEM...": IAF="integrate and fire"? Please specify.

line 185: eqn. 3: what is the index k? From other literature about IAF-TEMs, I would infer that it's the spike index? If that is correct, please also say that t_k is a spike time, and that these are strictly monotonically increasing with k etc. In fact, it would be very useful if there was a short introduction to IAF-TEMs: up to here, the paper only talks about continuous signals, now spikes are introduced (if I interpret everything correctly).

line 195-195: "Then there exists a number N ... can be perfectly recovered" this sounds like a sufficient condition. But in the proof of this theorem 1, line 224ff: "A necessary condition for the latter (solvability)...." so you only show a necessary condition? Hence, theorem 1 remains unproven, I think. In fact, I wonder what would happen if all filter kernels were the same? Then no matter how many neurons you added, you would not gain any further information, thus the signal recovery would not improve. To remedy this, I think the kernels would have to be constrained by some sort of linear independence condition, similar to [15].

figures 1,2,4: the v_i do not appear anywhere else in the paper. Please clarify their purpose.

section 5, examples. The examples are instructive. But I wonder why you did not choose a real video+audio for the examples -- it would have strengthened the connection to neuroscience
if you showed that your approach works on relevant datasets.

line 417: "...the first tractable computation model for multisensory integration..." wrong, see e.g. the works by Wei & Koerding, or Ernst & Banks. Please remove this sentence.










Summary: The extension of TEMs to the MIMO setting might be interesting for technical applications and decoding in Neuroscience. I believe Theorem 1 needs further clarification, but I am not entirely sure.

Submitted by Assigned_Reviewer_8

The authors model multisensory integration as a multiplexing process.

This paper is incredibly dense. I can find no fault with it's execution, but nor can I be sure that I follow it. What does seem clear, however, is that there is absolutely no connection to real biological systems, despite the claimed source of inspiration.
Summary: A seemingly thorough application, with questionable relevance or impact.
Author Feedback

Author rebuttal: We would like to thank reviewers for their comments about the manuscript.

We were astonished to see statements
(A)“there is absolutely no connection to real biological systems/Neuroscience”
(B)“the work is incremental and unlikely to have much impact”

(A):
Our work has irrefutable connections to biological systems/neuroscience as
(i) It addresses an experimental problem of jointly identifying receptive fields (RFs) of multisensory neurons [1] and employs biophysically grounded models of neurons to do so. Identification of multisensory RFs has not been possible since traditional methods require separate stimulus presentations for each modality. Importantly, joint (and not separate) stimulus presentation is often needed in experiments to elicit a response.
(ii) Multisensory integration has been observed in many cortical areas, including the superior colliculus, visual and auditory cortices. Precise rules by which neurons combine signals across modalities remain unknown. We provided tools for studying such rules within the spiking neuron framework.
(ii) RFs are well established in neuroscience. However, their proper estimation in circuits with spiking neurons is not. Studying multisensory integration by identifying multisensory RFs in cascade with biophysical neuron models is as biologically plausible as it gets.

(B):
(i) We present deep mathematical results addressing the problem of sensory integration at the level of individual neurons. The first major contribution was to work out a spiking model that integrates sensory stimuli in different spaces & dimensions. The second was to find conditions for inverting a nonlinear operator mapping multiple information streams into a single spike train. The third was to develop methods for identifying multisensory circuits.
(ii) Such problems have not been rigorously studied before. Works cited by reviewer #2 investigated multisensory integration in the context of psychophysical experiments, and not spiking neural circuits (i.e., no neural correlates).

We hope the rebuttal will clarify our approach/results and will motivate a further discussion between reviewers.

------ Assigned_Reviewer_6
“…although the method proposed does not seem to be biologically plausible form an algorithmic perspective...”
--> (A) When it comes to the decoding problem, by no means are we suggesting that a biological system implements a decoder. Rather, decoding algorithms can be used by researchers to (i) study neural processing and evaluate model performance [2,3]; (ii) design brain machine interfaces [4,5].
(B) Identification of neural circuits calls for finding parameters of phenomenological circuit models. Thus the question of biological plausibility from an algorithmic perspective simply does not arise in the identification setting. We presented algorithms that (i) provably identify RFs based on responses of spiking neuron models, including biophysical models, and (ii) work with both synthetic and naturalistic stimuli.

----- Assigned_Reviewer_7
“…the paper's most original contribution is the extensions of TEMs to the MIMO setting.”
-->The paper presents two results: (i) a circuit for multisensory processing (and not simply for multiple inputs and outputs) and (ii) identification methods for such circuits. The originality is in showing that
(a) the information about individual sensory stimuli (e.g., audio and video) can be decoded from the common pool of spikes. This is highly counterintuitive, given that spikes and spike-trains are not labeled. After all, audio and video have completely different dimensions and time scales;
(b) the identification problem for a single multisensory neuron is dual to multisensory encoding with a population of neurons. Without this duality, identification has been out of reach for systems/experimental neuroscientists -- until now.

“I wonder why you did not choose a real video+audio for the examples -- it would have strengthened the connection to neuroscience.”
-->Thank you for this suggestion. We can provide a new figure and a 6-second-long natural video with audio, if the paper is accepted.

“Theorem 1 needs to be corrected/clarified.”
--> We apologize for the confusion: part of the theorem was accidentally omitted. The sufficient condition is to have a large enough population of (different) neurons: N>=SUM_m[(2L_1+1)(2L_2+1)…(2L_{n_m-1} + 1)]. The necessary condition is that the total number of spikes is greater than
(2L_{n_m}+1)*SUM_m[(2L_1+1)(2L_2+1)…(2L_{n_m-1} + 1)] +N

“…what would happen if all filter kernels were the same? Then no matter how many neurons you added, you would not gain any further information.”
--> That is not entirely correct. For example, if all filters are Dirac-delta functions, the inputs are encoded directly without processing. However, one can recover all inputs if the neurons have different parameters (e.g., bias, capacitance, threshold of the IAF neuron). Imposing linear independence between kernels is another way to guarantee signal recovery.


----- Assigned_Reviewer_8
“…there is absolutely no connection to real biological systems, despite the claimed source of inspiration”
-->This assertion is beyond astonishing since we provide systems neuroscientists with (i) a detailed spiking neural circuit model for multisensory integration, (ii) a decoding algorithm to faithfully recover multisensory stimuli and (iii) an identification algorithm for multisensory systems. Identification of multisensory systems has not been even remotely possible due to the lack of a rigorous problem formulation. Our results are ahead of experimentation and suggest very specific experiments to anyone working in multisensory processing.

References:
[1] DC Kadunce, JW Vaughan, MT Wallace, BE Stein, 2001
[2] GB Stanley, FF Li, Y Dan, 1999
[3] MC-K Wu, SV David and JL Gallant, 2006
[4] BN Pasley et al, 2012
[5] TW Berger, VZ Marmarelis, SA Deadwyler, 2012